# Trends in the prevalence, incidence and surgical management of carpal tunnel syndrome between 1993 and 2013: an observational analysis of UK primary care records

Claire L Burton, Ying Chen, Linda S Chesterton, Danielle A van der Windt

Arthritis Research UK Primary Care Centre, Research Institute for Primary Care & Health Sciences, Keele University, Staffordshire, UK

**Correspondence to**
Dr Claire L Burton;
c.burton@keele.ac.uk

## ABSTRACT

**Objectives** To describe the prevalence, incidence and surgical management of carpal tunnel syndrome (CTS), between 1993 and 2013, as recorded in the Clinical Practice Research Datalink (CPRD).

**Design** We completed a series of cross-sectional epidemiological analyses to observe trends over time.

**Setting** Primary care data collected between 1993 and 2013, stored in the CPRD.

**Population** Individuals aged ≥18 years were selected. Prevalent and incident episodes of CTS and episodes of surgical intervention were identified using a list of preidentified Read codes.

**Analysis** We defined incident episodes as those with no preceding diagnostic code for CTS in the past 2 years of data. Episodes of surgery were expressed as a percentage of the prevalent population during the same calendar year. Joinpoint regression was used to determine significant changes in the underlying trend.

**Results** The prevalence of CTS increased over the study period, with a particular incline between 2000 and 2004 (annual percentage change 7.81). The female-to-male prevalence ratio reduced over time from 2.74 in 1993 to 1.93 in 2013. The median age of females and males with CTS were noted to increase from 49 and 53 years, respectively in 1993 to 54 and 59 years, respectively in 2013. Incidence was also noted to increase over time. After an initial increase between 1993 and 2007, the percentage of prevalent patients with a coded surgical episode began to decrease after 2007 to 27.41% in 2013 (annual percentage change –1.7).

**Conclusion** This study has demonstrated that the prevalence and incidence of CTS increased over the study period between 1993 and 2013. Rates of surgery for CTS also increased over the study period; however after 2007, the per cent of patients receiving surgery showed a statistically significant decline back to the rate seen in 2004.

## INTRODUCTION

Carpal tunnel syndrome (CTS) is a chronic focal compressive neuropathy caused by the entrapment of the median nerve at the level of the carpal tunnel in the wrist.[1] CTS is the most common presentation of the entrapment neuropathies[2] and is characterised by symptoms including paraesthesia, dysesthesia, sensory loss and eventually weakness and atrophy of the thenar muscle. Symptoms are usually localised to the hand but can spread proximally to the forearm, upper arm and even shoulder.[3] Despite causing relatively localised symptoms, CTS can have substantial physical, psychological and economic consequences.[4 5] In some cases, there may be associations with certain occupations (such as the care and leisure industry),[6] which involve the overuse of the hand and wrist as well as other physical comorbidities including pregnancy, diabetes, hypothyroidism and obesity.[7]

The diagnosis of CTS is generally accepted to be a clinical one (based on history and examination findings),[8] although electro-diagnostic tests are commonly requested to confirm the diagnosis or differentiate among diagnoses, especially in the presence of thenar atrophy and/or persistent numbness or if surgical management is being considered.[9] The treatment of CTS is usually defined as either surgical or conservative (non-surgical). Local steroid injections and night splinting form the mainstay of primary

care interventions in CTS, as indicated by national care pathways.[10 11] Patients with moderate, severe or deteriorating symptoms following conservative treatment or sudden and severe symptoms are recommended to be referred for consideration of surgery.[12] Carpal tunnel release surgery (CTR) is routinely carried out under local anaesthetic as day surgery. Open and endoscopic approaches are used to release the flexor retinaculum.[13] Previous studies have sought to estimate the prevalence and/or incidence of CTS. Such epidemiological studies have been diverse in their approach to the populations studied and case definitions applied.[14] The reported estimates for annual prevalence range from 3720 to 5700 per 100 000 per year[15–17] and the reported incidence from 72 to 8200 per 100 000 per year.[6 14 18–23] CTS is generally accepted to be more common in women; the female-to-male ratio ranges between 0.78 and 9.66.[14 15] A number of previous studies have observed the trends of prevalence or incidence over time and identified an increase,[19 20 24] with 2005 being the latest data collection point. The most recent primary care-based study in the UK used data between 1992 and 2000.[18]

### Strengths and limitations of this study

► Provides updated epidemiological data about a common and bothersome condition.
► Set in primary care, where most cases of carpal tunnel syndrome present.
► Uses a large primary care database, generalisable to the UK population.
► Relies on the correct coding and capture of episodes of carpal tunnel syndrome and carpal tunnel release surgery.

Episodes of CTR have also been shown to have increased, with audit data from one major tertiary UK Hand Centre suggesting that referral for CTR increased over a 10-year period from 59.7 to 112 per 100 000 population per year between 1989–1999 and 2000–2001.[25] Using Hospital Episode Statistics (HES) between 1998 and 2011, Bebbington and Furniss also observed an increase in the absolute number of patients with CTS and episodes of CTR; however, they also noted a decrease in the use of surgery post-2008.[26]

Previous studies have used a range of methods to classify episodes of CTS and have been conducted in a number of population settings. CTS is essentially a clinical diagnosis, and in the UK, the majority of patients will first present to and be managed within primary care. Only a proportion of these patients will be referred into more specialised services and since not all surgical episodes will take place in secondary care (hospitals), as community clinics are now receiving referrals, primary care records should capture the majority of episodes. Data from a high-quality source, representative of the UK population is necessary to support the planning and commissioning of services.

The aim of this study is therefore to provide updated estimates of the prevalence, incidence and surgical management of CTS and describe trends over a 20-year period, using data from a large national primary care database (Clinical Practice Research Datalink (CPRD)).

## METHODS

This was an observational study using the CPRD to estimate the prevalence, incidence and surgical management of CTS from 1993 to 2013. CPRD is a live, primary care database of anonymised medical records from general practices. It holds information of over 11.3 million patients from 674 practices in the UK since 1987; 4.4 million active

**Table 1** Read codes used to define a prevalent or incident episode of carpal tunnel syndrome

| Term | Read code |
|---|---|
| Carpal tunnel syndrome | F340 |
| Injection of carpal tunnel | 85BE.00 |
| Carpal tunnel release | 70560 |
| Endoscopic carpal tunnel release | 7056011 |
| Carpal tunnel decompression | 70564 |

(alive and currently registered) patients are currently contributing information to the datalink, which equates to 6.9% of the UK population.[27] The CPRD is broadly representative of the UK general population in terms of age, gender and ethnicity.[27] The CPRD has National Research Ethics Committee approval for observational research using primary care data and as such no further permissions were required. The Independent Scientific Advisory Committee study protocol 14_167 was approved in September 2014.

During clinical interactions, Read codes are used to record signs and symptoms, treatments and therapies, investigations, occupations, diagnoses and appliances. Read codes make up a hierarchical 'thesaurus' stored by the computer. Clinical information is hence stored in a retrievable and analysable format.[28]

The study population consisted of men and women over 18 years of age. Data was used from practices which met a data quality standard based on continuity of recorded data, and from patients who had a record including at least their registration status, age and gender. These quality standards were required to have been met for at least 2 years prior to an incident episode and at the point of diagnosis for a prevalent episode.[27]

Prevalent and incident patients were identified by a consultation recorded using one of the Read codes listed in table 1. Some treatment codes and in the case of in injections, linked prescription data, were included as evidence of diagnosis as per previous studies.[18] Pilot work using a local primary care database (Consultations in Primary Care Archive (CiPCA)[29]) had noted that 30% of CTS cases with a treatment code (ie, CTR or a coded carpal tunnel injection) had not initially received a diagnosis code. This means that at presentation, patients may have been attributed a more generic term such as 'hand pain' and later gone on to receive condition-specific treatment. Hence, treatment codes were used to capture such cases, which would be missed when using diagnostic codes only.

The prevalence of individuals consulting with CTS was calculated per annum. The numerator for prevalence was the number of patients with a record of a CTS diagnosis or evidence of an episode of CTR or a carpal tunnel injection (CTI), in each calendar year. In order to determine annual incidence, the numerator was the

number of patients with a record of CTS or evidence of CTR or CTI, without a prior record of these codes during a run-in period of 2 years. This 2-year run-in period was based on expert consensus with the aim of estimating the period of time during which a new episode of CTS may develop. It was felt unlikely that a patient with ongoing bothersome symptoms would not have presented in primary care within this 2-year period. This however is an assumption made in order to define incident cases in this data set. It remains possible that patients had CTS in the community and did not present, presented in an alternative setting or indeed had a misdiagnosis/uncoded diagnosis made. CTS could present as a new episode in the contralateral wrist sometime after the initial presentation, hence it was not felt possible to define this criterion as 'no previous recorded episode'. All incidence patients were therefore required to have complete registration for this two calendar years prior to the event date. Pilot work in CiPCA had shown that over 9 years observed, 4% of potential incident cases were lost due to the lack of 2 years registration data required to define an incident episode.

The denominator population for calculation of prevalence was the total up-to-standard person-years contributed to CPRD by patients over the age of 18 years, for each annual period between 1993 and 2013. In order to apply the same criteria to both the numerator and denominator populations, the denominator populations for calculating incidence were also required to have registration at the mid-point of the year, two calendar years before the index year.

Episodes of CTR were identified using Read codes as shown in table 2. In addition, codes used to define 'rerelease of carpal tunnel' and 'revision of carpal tunnel release' were included as a surgical episode (if first recorded). These terms were not included in the definition of CTS for the estimation of prevalence and incidence as they may not have indicated an episode of 'idiopathic' CTS but rather iatrogenic symptoms following previous (unsuccessful) surgery. Of note, revision codes contributed 1.00% of the total surgical codes used. Results were expressed as the percentage of patients with a prevalent episode of CTS having a code of CTR in the same calendar year. Percentages were calculated based on the number of prevalent cases as opposed to incident cases as

| Table 2 | Read codes used to define a surgical episode |
|---|---|
| **Term** | **Read code** |
| Carpal tunnel release | 817 |
| Rerelease of carpal tunnel | 16896 |
| Endoscopic carpal tunnel release | 39335 |
| Revision of carpal tunnel release | 97195 |
| Carpal tunnel decompression | 19249 |

it was felt likely that patients would receive surgery in the annual period following their index consultation.

## Statistical methods

Age-specific and sex-specific annual prevalence and incidence were determined for each calendar year, between 1993 and 2013 and presented as n/10 000 person-years. For CI calculation a Poisson distribution was used. As a sensitivity analysis, age-standardised and sex-standardised annual figures of CTS prevalence and incidence for each year were also calculated, using population estimates provided by the website of the Office of National Statistics.[30] Unstandardised and standardised rates were very similar, hence we report unstandardised rates as the primary outcome. The age-standardised and sex-standardised estimates of the annual prevalence and incidence of CTS are shown in online supplementary table 1.

Episodes of CTR were identified and the frequency in each calendar year expressed as a percentage of the prevalent population for the same time period. Emerging trends were described. Joinpoint regression was used to determine mean annual percentage change (APC) and assess when significant changes ('Joinpoints') occurred in the underlying trend for incidence, prevalence and surgery. This method assists the exploration of the potential influence of changes in practice, although such potential associations cannot be proven.[31 32] Models were fitted using the Joinpoint Regression Program (V.4.3.1.0) and the best fitting model chosen (up to five Joinpoints).

## Patient and public involvement

Patients were not directly involved in the design of this study; however, the results will be used to inform discussions regarding further research in this field with our local Research User Group.

## RESULTS
### Trends in prevalence

Table 3 presents the prevalence (crude estimates) of patients presenting in primary care with CTS between 1993 and 2013 and the demographics of the population. The denominator population for prevalence increased from 1 117 433 person-years in 1993 to 3 473 094 person-years in 2013. The total prevalence in 1993 was 26.03 per 10 000 person-years (95% CI 25.10 to 27.00), and for 2013, 36.08 per 10 000 person-years (95% CI 35.45 to 36.72). As shown in figure 1 and corresponding table 4, prevalence appeared to decrease between 1993 and 2000 (APC=−0.8%, 95% CI −2.6 to 1.0). It then increased between 2000 and 2004 (APC=7.8%, 95% CI 3.1 to 12.7) and then increased at a slower rate between 2004 and 2013 (APC=1.1%, 95% CI 0.4 to 1.8). The female-to-male ratio reduced over time from 2.74 in 1993 to 1.93 in 2013. The median age of female and male patients with CTS increased from 49 and 53 years, respectively in 1993 to 54 and 59 years, respectively in 2013 (see online supplementary table 2). Online supplementary table 3 and

**Table 3** Crude prevalence of carpal tunnel syndrome (n/10 000 person-years) per calendar year, as presented in UK primary care (Clinical Practice Research Datalink)

| Year | Number of person-years | Prevalent individuals | Total crude prevalence per 10000 person-years, (95% CI) | Female prevalence per 10000 person-years, (95% CI) | Male prevalence per 10000 person-years, (95% CI) | Female:male |
|---|---|---|---|---|---|---|
| 1993 | 1 117 443 | 2909 | 26.03 (25.10 to 27.00) | 37.52 (35.96 to 39.13) | 13.69 (12.72 to 14.71) | 2.74 |
| 1994 | 1 198 256 | 3188 | 26.61 (25.69 to 27.55) | 37.23 (35.73 to 38.79) | 15.21 (14.23 to 16.25) | 2.45 |
| 1995 | 1 286 800 | 3343 | 25.98 (25.11 to 26.88) | 36.64 (35.20 to 38.12) | 14.58 (13.65 to 15.56) | 2.51 |
| 1996 | 1 437 567 | 3706 | 25.78 (24.96 to 26.62) | 36.75 (35.38 to 38.16) | 14.09 (13.23 to 15.00) | 2.61 |
| 1997 | 1 681 756 | 4190 | 24.91 (24.17 to 25.68) | 34.87 (33.64 to 36.14) | 14.34 (13.53 to 15.18) | 2.43 |
| 1998 | 1 899 393 | 4884 | 25.71 (25.00 to 26.45) | 36.57 (35.38 to 37.79) | 14.22 (13.46 to 15.01) | 2.57 |
| 1999 | 2 289 158 | 5696 | 24.88 (24.24 to 25.54) | 35.21 (34.14 to 36.30) | 14.01 (13.32 to 14.72) | 2.52 |
| 2000 | 2 787 457 | 6998 | 25.11 (24.52 to 25.70) | 34.82 (33.86 to 35.81) | 14.90 (14.26 to 15.57) | 2.34 |
| 2001 | 3 057 458 | 8137 | 26.61 (26.04 to 27.20) | 36.46 (35.52 to 37.42) | 16.31 (15.67 to 16.98) | 2.23 |
| 2002 | 3 385 511 | 9722 | 28.72 (28.15 to 29.29) | 39.33 (38.40 to 40.28) | 17.64 (17.00 to 18.29) | 2.23 |
| 2003 | 3 552 908 | 11124 | 31.31 (30.73 to 31.90) | 43.61 (42.66 to 44.59) | 18.53 (17.90 to 19.18) | 2.35 |
| 2004 | 3 712 172 | 12622 | 34.00 (33.41 to 34.60) | 47.20 (46.23 to 48.19) | 20.33 (19.68 to 20.99) | 2.32 |
| 2005 | 3 808 183 | 12741 | 33.46 (32.88 to 34.04) | 46.37 (45.42 to 47.34) | 20.09 (19.45 to 20.74) | 2.31 |
| 2006 | 3 857 487 | 12718 | 32.97 (32.40 to 33.55) | 45.82 (44.88 to 46.78) | 19.69 (19.07 to 20.33) | 2.33 |
| 2007 | 3 904 068 | 13222 | 33.87 (33.29 to 34.45) | 46.35 (45.41 to 47.31) | 20.99 (20.35 to 21.65) | 2.21 |
| 2008 | 3 897 624 | 14030 | 36.00 (35.40 to 36.60) | 49.12 (48.15 to 50.11) | 22.46 (21.79 to 23.14) | 2.19 |
| 2009 | 3 894 989 | 14500 | 37.23 (36.60 to 37.81) | 50.68 (49.69 to 51.68) | 23.35 (22.68 to 24.05) | 2.17 |
| 2010 | 3 842 773 | 14166 | 36.86 (36.26 to 37.48) | 49.75 (48.76 to 50.75) | 23.57 (22.88 to 24.27) | 2.11 |
| 2011 | 3 769 676 | 13529 | 35.89 (35.29 to 36.50) | 47.98 (47.00 to 48.97) | 23.36 (22.67 to 24.07) | 2.05 |
| 2012 | 3 714 877 | 13388 | 36.04 (35.43 to 36.66) | 47.57 (46.59 to 48.56) | 24.05 (23.35 to 24.78) | 1.98 |
| 2013 | 3 473 094 | 12532 | 36.08 (35.45 to 36.72) | 47.19 (46.18 to 48.21) | 24.49 (23.75 to 25.25) | 1.93 |

supplementary figures 1 and 2 further illustrate the crude prevalence of CTS over time by age and gender. The prevalence of CTS appears to increase with age in the male population, whereas the prevalence in women peaks in the 50–59 years age group, dips in the 60–69 years age group and then peaks once more in the 70+ years age group.

### Trends in incidence

Table 5 presents the annual incidence (crude estimates) for patients presenting in UK primary care with carpal tunnel syndrome between 1993 and 2013 and the demographics of the population. The denominator population for incidence, which is dependent on patients having 2 years up to standard data prior to the mid-point of the year in question, increased from 783 330 person-years in 1993 to 3 015 670 person-years in 2013. The crude incidence in 1993 was 20.22 per 10 000 person-years (95% CI 19.24 to 21.24), and for 2013, 27.68 per 10 000 person-years (95% CI 27.09 to 28.28). As shown in figure 2 and table 6, the results of the best fitting Joinpoint regression suggest the incidence increased between 1993 and 2000 (APC=0.3, 95% CI −2.3 to 2.9). It then increased more quickly between 2000 and 2004 (APC=6.9, 95% CI 0.5 to 13.7), before slowing between 2004 and 2013 (APC=0.7. 95% CI −0.2 to 1.6). The female-to-male ratio reduced

over time from 2.57 in 1993 to 1.88 in 2013. The median age of female and male patients were noted to increase from 50 and 51 years, respectively in 1993 to 55 and 59 years, respectively in 2013 (see online supplementary table 4). See online supplementary table 5 and supplementary figures 3 and 4 further illustrate the incidence of CTS over time by age and gender. As with prevalence, the incidence of CTS appears to increase with age in the male population, whereas the prevalence in women peaks in the 50–59 years age group, dip in the 60–69 years age group and then peak once more in the 70+ years age group.

### Trends in the percentage of patients with carpal tunnel syndrome receiving surgical management

Table 7 presents the percentage of prevalent patients with a recorded episode of CTR in each calendar year between 1993 and 2013 and the demographics of this sample. The percentage of all patients with a recorded episode of CTR in 1993 was 19.35%, and for 2013, 27.41%. As shown in figure 3 and corresponding table 8, the percentage of patients with a coded episode of CTR increased between 1993 and 2007 (APC=2.6, 95% CI 1.9 to 3.2). It then appeared to decrease between 2007 and 2013 (APC=−1.7, 95% CI −3.3 to −0.3). The median age of females and males receiving CTR were noted to increase from 53 and

 Burton CL, *et al*. BMJ Open 2018;**8**:e020166. doi:10.1136/bmjopen-2017-020166

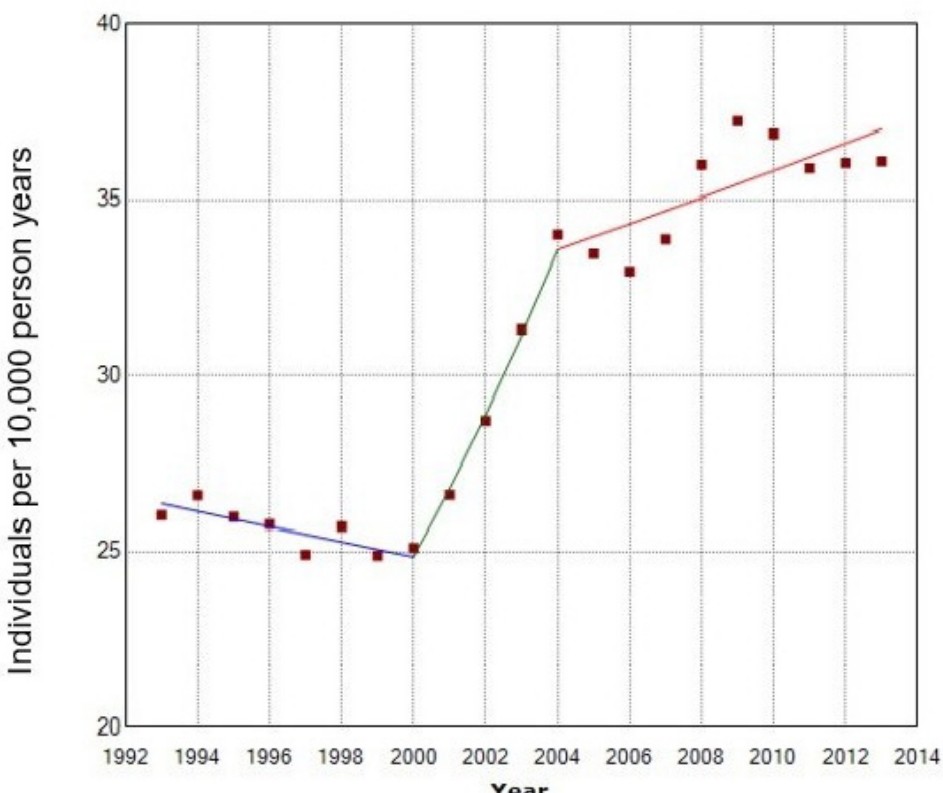

**Figure 1** Joinpoint analysis of the crude prevalence of carpal tunnel syndrome between 1993 and 2013. ^, reflects significance at the 0.05 level; APC, annual percentage change.

55 years, respectively in 1993 to 57 and 62 years, respectively in 2013.

## DISCUSSION

While the prevalence and incidence of CTS have increased over the study period 1993–2013, results show that episodes of surgery, increased until 2007 and declined thereafter.

Online supplementary tables 6 and 7 summarise estimates of the prevalence, incidence and sex ratios of CTS from a previous scoping review of literature pertaining to the general population, demonstrating the substantial variation in results between studies, which may partly be the results of differences in definition of CTS applied and population observed. Studies which also used primary care data showed a similar estimate of the incidence of CTS in a UK primary care population[18] and similarly

reported an increase in incidence over time, although in a Dutch primary care population.[21] As described in previous studies, CTS shows a peak in prevalence and incidence in women of middle age (50–59 years age group, likely due to hormonal changes around the time of the menopause),[18] while in the male population, the prevalence and incidence of CTS increased with age. Gelfman et al also commented that an increasing number of older people presenting with CTS had been noted over the course of their study.[20] The increase in the prevalence and incidence of CTS in the older-aged male groups, may partially account for the observed decrease in the female-to-male ratio, over time.

The variability in the case definition of CTS was highlighted by Descatha et al,[33] who identified seven case definitions of CTS proposed for use in population-based studies. Definitions included variations of symptoms

| | | | Annual | | | Test | |
|---|---|---|---|---|---|---|---|
| Segment | Lower end point | Upper end point | percentage change | Lower 95th CI | Upper 95th CI | statistic (t) | Prob > \|t\| |
| 1 | 1993 | 2000 | −0.8 | −2.6 | 1.0 | −1.0 | 0.3 |
| 2 | 2000 | 2004 | 7.8* | 3.1 | 12.7 | 3.7 | 0.0 |
| 3 | 2004 | 2013 | 1.1* | 0.4 | 1.8 | 3.4 | 0.0 |

**Table 4** Joinpoint analysis of crude prevalence

*Reflects significance at the 0.05 level.

**Table 5** Crude incidence of carpal tunnel syndrome (n/10 000 person-years) per calendar year, as presented in UK primary care (Clinical Practice Research Datalink)

| Year | Number of person-years | Incident individuals | Total crude incidence per 10 000 person-years (95% CI) | Female incidence per 10 000 person-years (95% CI) | Male incidence per 10 000 person-years (95% CI) | Female:male |
|------|------|------|------|------|------|------|
| 1993 | 783 330 | 1584 | 20.22 (19.24 to 21.24) | 28.72 (27.09 to 30.42) | 11.17 (10.14 to 12.29) | 2.57 |
| 1994 | 868 616 | 1797 | 20.69 (19.74 to 21.67) | 28.52 (26.97 to 30.13) | 12.38 (11.34 to 13.69) | 2.30 |
| 1995 | 1 003 593 | 1963 | 19.56 (18.70 to 20.45) | 27.53 (26.12 to 29.00) | 11.12 (10.20 to 12.10) | 2.48 |
| 1996 | 1 065 068 | 2142 | 20.11 (19.27 to 20.98) | 28.39 (27.00 to 29.84) | 11.37 (10.47 to 12.33) | 2.50 |
| 1997 | 1 150 299 | 2306 | 20.05 (19.24 to 20.88) | 28.39 (27.05 to 29.79) | 11.25 (10.39 to 12.16) | 2.52 |
| 1998 | 1 300 074 | 2696 | 20.74 (19.95 to 21.52) | 29.65 (28.57 to 31.22) | 11.37 (10.56 to 12.23) | 2.61 |
| 1999 | 1 497 673 | 3030 | 20.23 (19.52 to 20.10) | 28.53 (27.35 to 29.75) | 11.54 (10.77 to 12.34) | 2.47 |
| 2000 | 1 682 027 | 3462 | 20.58 (19.90 to 21.28) | 28.66 (27.54 to 29.81) | 12.15 (11.41 to 12.93) | 2.36 |
| 2001 | 2 019 596 | 4391 | 21.74 (21.10 to 22.40) | 29.72 (28.68 to 30.79) | 13.46 (12.74 to 14.20) | 2.21 |
| 2002 | 2 456 761 | 5718 | 23.27 (22.68 to 31.78) | 31.78 (30.78 to 32.79) | 14.47 (13.80 to 15.17) | 2.20 |
| 2003 | 2 669 111 | 6772 | 25.37 (24.77 to 25.98) | 35.13 (34.14 to 36.14) | 15.33 (14.67 to 16.02) | 2.29 |
| 2004 | 2 779 821 | 7868 | 28.30 (27.68 to 28.94) | 39.22 (38.19 to 40.27) | 17.10 (16.42 to 17.81) | 2.29 |
| 2005 | 3 164 506 | 8113 | 25.64 (25.08 to 26.20) | 35.55 (34.63 to 36.48) | 15.49 (14.88 to 16.12) | 2.30 |
| 2006 | 3 307 051 | 8337 | 25.21 (24.67 to 25.76) | 34.91 (34.02 to 35.82) | 15.27 (14.68 to 15.89) | 2.29 |
| 2007 | 3 343 009 | 8865 | 26.52 (25.97 to 27.08) | 35.76 (34.86 to 36.67) | 17.07 (16.45 to 17.71) | 2.09 |
| 2008 | 3 341 299 | 9437 | 28.24 (27.68 to 28.82) | 38.23 (37.30 to 39.17) | 18.06 (17.42 to 18.72) | 2.12 |
| 2009 | 3 383 196 | 9918 | 29.32 (28.74 to 29.90) | 39.73 (38.79 to 50.68) | 18.69 (18.04 to 19.36) | 2.13 |
| 2010 | 3 357 338 | 9634 | 28.70 (28.13 to 29.27) | 38.70 (37.77 to 39.64) | 18.46 (17.82 to 19.13) | 2.10 |
| 2011 | 3 269 296 | 9083 | 27.78 (27.21 to 28.36) | 37.11 (36.19 to 38.05) | 18.20 (17.54 to 18.87) | 2.04 |
| 2012 | 3 222 880 | 9011 | 27.96 (27.39 to 28.54) | 36.44 (35.52 to 37.88) | 19.23 (18.56 to 19.93) | 1.89 |
| 2013 | 3 015 670 | 8346 | 27.68 (27.09 to 28.28) | 35.95 (35.01 to 36.92) | 19.12 (18.43 to 19.84) | 1.88 |

only; symptoms and examination findings; symptoms and either physical examination or electrodiagnostic results and symptoms and electrodiagnostic results. This study showed a range in the population prevalence of CTS from 2.5% to 11%, with studies using less specific case definitions yielding higher prevalence rates.[33] Misclassification ranged between 1% and 10%. The prevalence of CTS in any given population is likely therefore to depend on the definition of CTS applied. The case definition in our study is derived from general practitioner (GP)-recorded diagnosis and treatment codes, which may have been based on clinical findings alone; those who have had further investigations and those who have received definitive condition-specific treatment. Hence, it uses a pragmatic approach, across a large population that will include all patients presenting to their GP with symptoms. Our study methods do however assume that patients with symptoms will be presenting in primary care or be receiving definitive coded treatment. The study will not capture patients with chronic symptoms who are not presenting in primary care or who had a coded episode of surgery or injection.

Although Joinpoint analysis does not provide evidence for the cause of a change in observed outcomes, it highlights when a significant change in trend has taken place.

Our results suggest that the annual percentage change in prevalence and incidence was highest between 2000 and 2004. A possible reason for this may be the publication of the UK Government's information technology strategy for the NHS in 1998,[34] which proposed that by 2005, the person-based electronic health record (HER), would have been fully implemented.[35] Although no direct evidence for this was found, it may be possible that with the increasing use of IT systems in primary care and attention to providing Read codes for each consultation, episodes of CTS were more frequently and accurately recorded. This would not however explain the continuing increase of the incidence in CTS post-2005.

Between 2000 and 2004, the Government implemented the second phase of its 'War on Waiting', that is, the reduction of waiting times. For example, the maximum wait for a day-case procedure (eg, a CTR) was reduced from 18 months to 6 months.[36] The peak in prevalence of CTS (with our definition partly based also on treatment codes, which in 2013 constituted 29.36% of prevalent patients) observed in 2004 may therefore be partly explained by the fact that patients requiring surgery were 'accumulating' between 2000 and 2004 and subsequently received definitive treatment. This effect would however not be expected to impact so heavily on the incidence, which

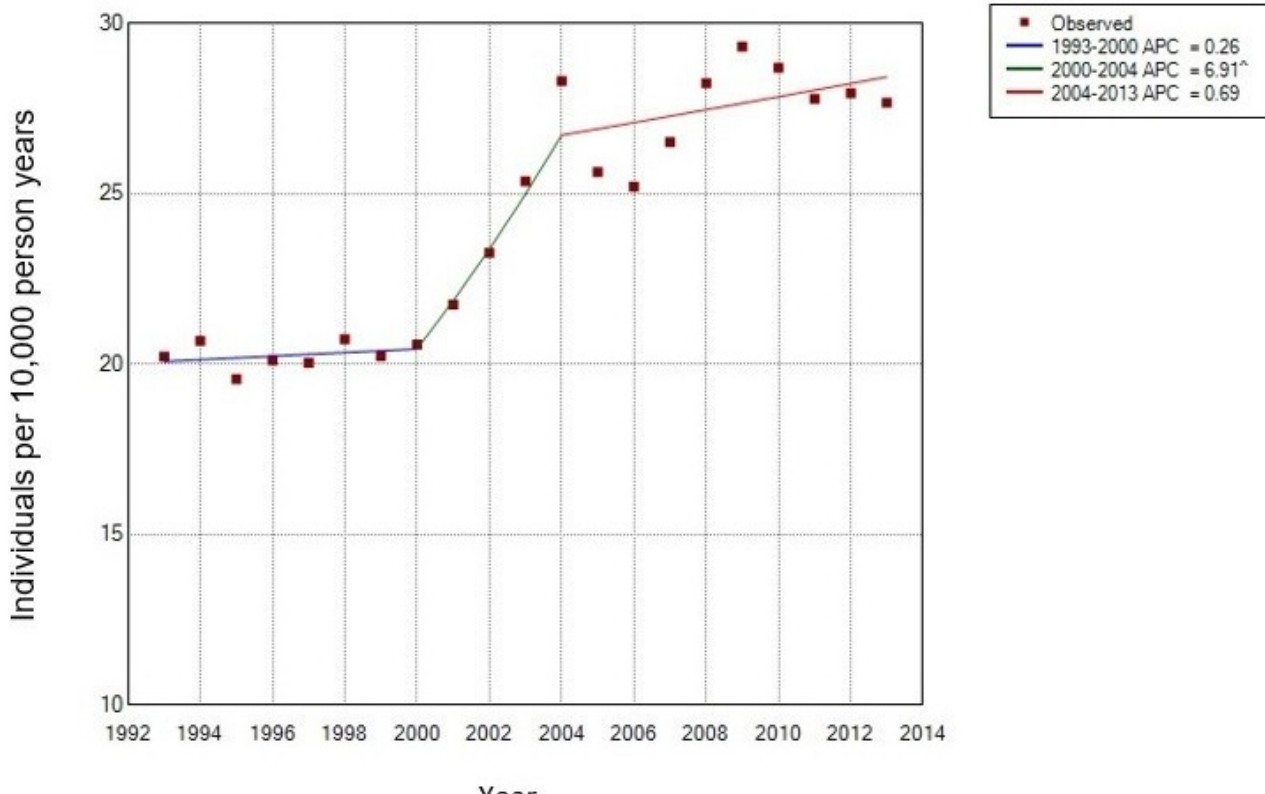

**Figure 2** Joinpoint analysis of the crude incidence of carpal tunnel syndrome between 1993 and 2013. ^, reflects significance at the 0.05 level; APC, annual percentage change.

disregards repeat patient presentations in subsequent annual periods, unless patients with a less specific code received treatment and appeared as an incident case. The introduction of the 18-week target of time from referral to treatment in 2008 did not seem to have a similar impact on estimates of prevalence or incidence of CTS, which makes it less certain to what extent these policy changes may have influenced our results. There are likely to be further reasons behind the observed changes.

The change in trends of 2004 may also represent a change in service. The introduction of the Quality and Outcome Framework (QOF) occurred with the advent of the General Medical Services contract in 2004. Although there has never been a musculoskeletal health domain, the importance of coding to maintain registers and evidence of outcomes in line with QOF may have influenced coding behaviour.

At the same time as QOF, Primary Care Trusts (PCTs) were given a role in commissioning services. The ability of PCTs to commission new services heralded the development of the Musculoskeletal Interface Clinics (MIC), which act as a 'one stop shop' for patients with musculoskeletal problems. A referral to this clinic from primary care may also be a reason prevalent patients with persisting symptoms stopped presenting in primary care.

These three factors (improved coding, service redevelopment and a reduction in waiting times) may all partly explain the change in incidence and prevalence of CTS between 2000 and 2004 but are unlikely to fully explain the observed trends. Further factors of potential influence may include the increasing rates of risk factors of CTS such as diabetes and obesity.[37 38] While standardising the prevalence and incidence by age and gender did not change the overall picture of the changing trends, online

| | | | Annual | | | Test | |
| Segment | Lower end point | Upper end point | percentage change | Lower 95th CI | Upper 95th CI | statistic (t) | Prob > \|t\| |
|---|---|---|---|---|---|---|---|
| 1 | 1993 | 2000 | 0.3 | −2.3 | 2.9 | 0.2 | 0.8 |
| 2 | 2000 | 2004 | 6.9* | 0.5 | 13.7 | 2.3 | 0.0 |
| 3 | 2004 | 2013 | 0.7 | −0.2 | 1.6 | 1.7 | 0.1 |

**Table 6** Joinpoint analysis of crude incidence

*Reflects significance at the 0.05 level.

**Table 7** Percentage of patients with carpal tunnel syndrome with a recorded episode of carpal tunnel release surgery per calendar year, as presented in UK primary care (Clinical Practice Research Datalink)

| Year | Episodes per 10 000 person-years | % prevalent individuals having surgery | % prevalent females having surgery | % prevalent males having surgery | Female median age (25%–75% IOR) | Male median age (25%–75% IQR) |
|------|------|------|------|------|------|------|
| 1993 | 5.04 | 19.35 | 18.78 | 21.03 | 53 (43–64) | 55 (44–69) |
| 1994 | 5.70 | 21.42 | 20.62 | 23.52 | 53 (43–68) | 58 (45–70) |
| 1995 | 6.19 | 23.81 | 23.40 | 24.92 | 53 (42–67) | 55 (44–70) |
| 1996 | 5.41 | 20.99 | 20.48 | 22.43 | 53 (44–65) | 52 (40–65) |
| 1997 | 5.70 | 22.89 | 22.14 | 24.81 | 53 (45–67) | 56 (42–69) |
| 1998 | 5.73 | 22.28 | 21.28 | 25.00 | 53 (44–65) | 53 (44–65) |
| 1999 | 6.24 | 25.09 | 24.60 | 26.38 | 54 (44–67) | 56 (46–70) |
| 2000 | 6.41 | 25.54 | 24.84 | 27.23 | 54 (44–68) | 56 (45–69) |
| 2001 | 6.88 | 25.87 | 25.95 | 25.68 | 55 (45–68) | 58 (46–71) |
| 2002 | 7.02 | 24.46 | 24.19 | 25.09 | 57 (46–71) | 55 (45–68) |
| 2003 | 8.26 | 26.39 | 25.88 | 27.66 | 56 (45–67) | 57 (46–71) |
| 2004 | 9.34 | 27.48 | 27.38 | 27.74 | 56 (46–67) | 57 (47–68) |
| 2005 | 9.70 | 29.00 | 28.31 | 30.65 | 57 (47–68) | 58 (46–71) |
| 2006 | 9.36 | 28.40 | 28.31 | 28.61 | 57 (47–68) | 60 (48–72) |
| 2007 | 9.71 | 28.66 | 28.26 | 29.59 | 56 (46–69) | 59 (48–71) |
| 2008 | 10.53 | 29.25 | 29.00 | 29.82 | 56 (46–68) | 60 (49–72) |
| 2009 | 10.92 | 29.32 | 28.73 | 30.66 | 56 (46–70) | 61 (49–72) |
| 2010 | 10.40 | 28.22 | 27.57 | 29.62 | 57 (47–71) | 61 (48–73) |
| 2011 | 9.47 | 26.37 | 26.11 | 26.93 | 57 (47–70) | 61 (49–73) |
| 2012 | 9.48 | 26.31 | 25.89 | 27.19 | 57 (47–71) | 60 (49–73) |
| 2013 | 9.89 | 27.41 | 26.47 | 29.30 | 57 (48–70) | 62 (51–74) |

supplementary figure 1 suggests that the prevalence of CTS increased most obviously in the male and female over 70 years age groups.

The Joinpoint analysis suggested an increase in surgical management of CTS between 1993 and 2007 (APC=2.55), followed by a reducing trend between 2007 (95% CI 2004 to 2009) and the end of the study in 2013 (APC=−1.72).

Previous studies have described the epidemiology and the rates of CTR in the UK. This study provides updated data observing the presenting primary care population. Using data from the General Practice Research Database (GPRD) (forerunner to CPRD), Latinovic et al reported that 31% of patients with CTS had surgery in 2000,[18] which is similar to the 25.5% found in our study at the same time point. The small difference between the estimates may be the result of a difference in the calculation used to derive the denominator population. Audit data from one tertiary hand centre, Wildin et al also showed that the rate of referrals for CTR surgery had increased over the 10 years between 1989–1999 and 2000–2001.[25] Furthermore, Bebbington and Furniss observed demographic population shifts in hand conditions including CTS within HES, which record diagnoses and procedures performed within the National Health Service (NHS) Hospitals in England. They used linear regression to predict future trends in hand surgery, showing that while absolute numbers of CTS diagnoses and CTR procedures increased between 1998 and 2011, the pre-2008 increase in CTR was significantly steeper than the post-2008 slope (p<0.001).[26] This is suggestive of a decrease in the surgical management of CTS in terms of the proportion of patients with CTS having an operation, but not necessarily in the numbers of surgical episodes in absolute terms, which Bebbington and Furniss predict will have increased by 99% (95% CI 65 to 132) in 2030 compared with 2011.[26] The data from CPRD however, suggested a reduction in both real-term episodes of CTR as well as the proportion of the (increasing) prevalent population receiving surgical treatment.

We may speculate regarding potential reasons for the initial increase in surgical management of CTS, for example, increased access to specialist services (eg, community-based MIC), increased litigation leading to more definitive treatments being sought and increased patient expectations and demand, but we have no evidence for such explanations.

The decreasing trend in the use of CTR post-2007 is likely to be multifactorial; however, the changing structure of the NHS and its funding streams may have influenced the observed trend. Around 2007–2008, practice-based

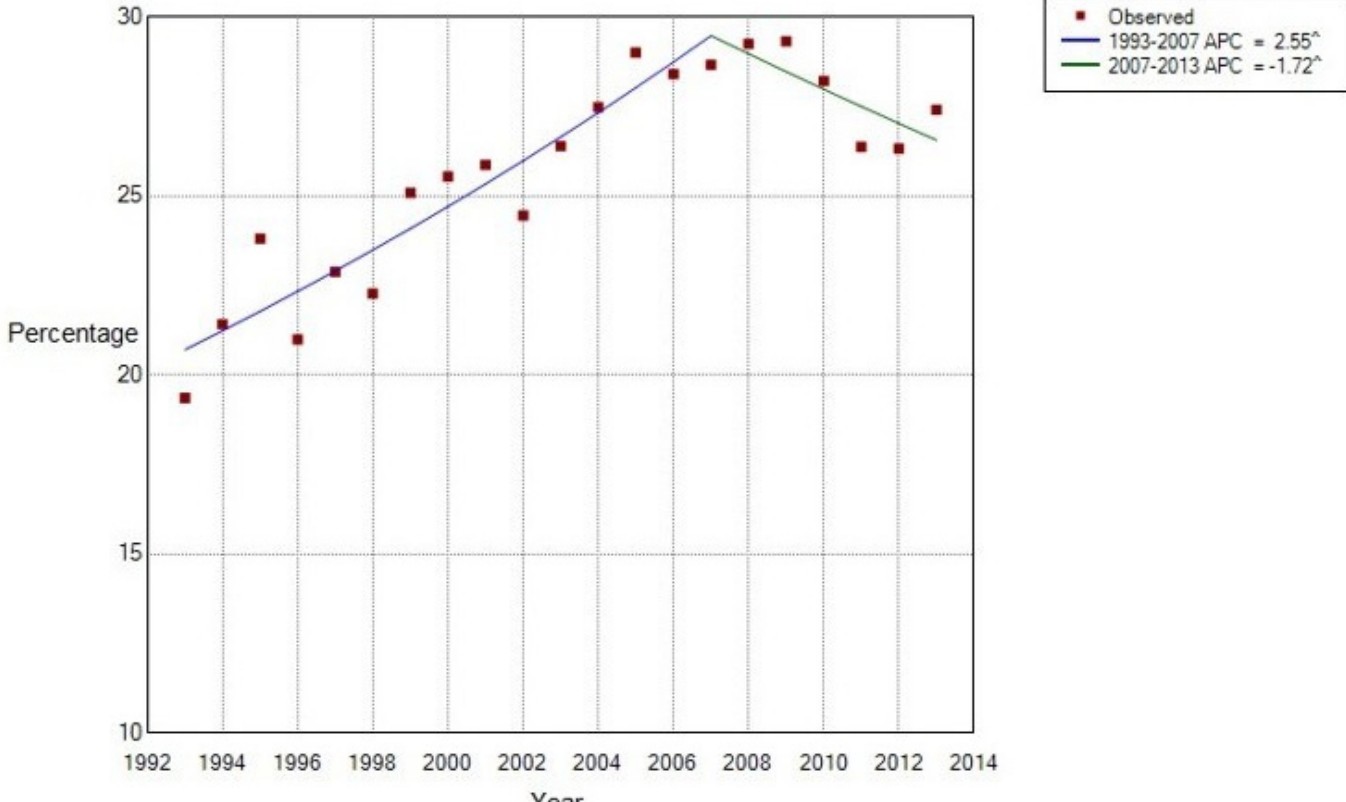

**Figure 3** Joinpoint analysis of the percentage of prevalent patients with a recorded episode of carpal tunnel release, in each calendar year syndrome, between 1993 and 2013. ^, reflects significance at the 0.05 level; APC, annual percentage change.

commissioning was being introduced. This gave primary care notional budgets with which to purchase care for their patients with the aim of aligning clinical and financial responsibility. Restricting access to certain procedures including CTR, by implementing prespecified criteria, was one way to help achieve this, which may have resulted in a reduction in the use of CTR.

There are a number of limitations associated with the data in this study. The accuracy of consultation data is dependent on the validity of the computerised information it uses. In a review of 212 publications which aimed to validate diagnoses recorded in GPRD data, Herrett *et al* reported that the median proportion of cases with a confirmed diagnosis was 89% (range 24%–100%), but the majority of publications did not present the sensitivity of a coded diagnosis, which means that information regarding the proportion of missed cases is lacking. Potential misclassification; non-attendance

in primary care; variation in between GP coding and a lack of coding may all lead to an unmeasured shortfall in observed cases.[27 39] This study relies on the diagnosis of CTS to be correct and the subsequent coding to be precise. While CTS diagnoses have not been validated, in a study comparing musculoskeletal diagnoses in four different databases, Jordan *et al* suggested that musculoskeletal coding in GPRD was less reliable than in its other healthcare datasets including CiPCA.[40] We took measures to reduce the effect of miscoding (eg, including surgery and injection codes in prevalence measures, if diagnostic codes had not been used), but it is possible that results will not be entirely representative of the true prevalence and incidence of CTS.

Given the lack of clarity in the accuracy of coding and the likelihood that associated clinical encounters following a CTR were coded using a surgical code, only the first surgical code could reliably be used to indicate an

**Table 8** Joinpoint analysis of the use of surgery

| Segment | Lower end point | Upper end point | Annual percentage change | Lower 95th CI | Upper 95thCI | Test statistic (t) | Prob > \|t\| |
|---|---|---|---|---|---|---|---|
| 1 | 1993 | 2007 | 2.6* | 1.9 | 3.2 | 8.2 | 0.0 |
| 2 | 2007 | 2013 | −1.7* | −3.1 | −0.3 | −2.6 | 0.0 |

*Reflects significance at the 0.05 level.

episode of surgery. This is likely to have led to an underestimation of surgical episodes being identified as episodes on the contralateral hand will have been automatically discounted as they were undistinguishable. Furthermore, prevalence and incidence were similarly likely to have been underestimated as repeat presentations for the ipsilateral hand are indistinguishable from presentations in the contralateral hand.

While CPRD provides a large generalisable sample, which has substantial benefits when estimating epidemiological trends, it cannot directly measure patient-reported outcomes. Furthermore, surgery can be seen as a 'gold standard' treatment, but it does not necessarily signify cure. A review of the surgical treatment of CTS reported that 70%–90% of patients undergoing a CTR have a good outcome (definitions varied).[41] In a retrospective cohort study over a mean follow-up of 13 years postsurgery, 88% of patients were either completely satisfied or very satisfied with surgery. Seventy-four per cent reported their symptoms had completely resolved; 1.8% (113 patients) had undergone repeat surgery.[42] There is little evidence however that CTR is an appropriate initial management option for patients presenting to primary care with mild-to-moderate symptoms, especially in the absence of high-quality trial evidence that conservative management is ineffective.[43 44]

Future research in this field could describe the characteristics of patients presenting with CTS in greater detail, and observe course and prognosis of CTS in primary care. It may then be possible to identify predictors of the outcome of primary care management, and potentially identify patients requiring surgery.

## CONCLUSION

An increase in the incidence and prevalence of CTS is likely to lead to an increased demand on services and cost to the healthcare economy.[26] This study has demonstrated an increase in the prevalence and incidence of physician diagnosed CTS over the study period between 1993 and 2013. Rates of referral for CTS and surgical intervention have also increased over the study period; however in the later years of the study, the per cent of patients receiving surgery has begun to decline .

**Acknowledgements** The authors wish to thank Dr Dahai Yu for his assistance with the CPRD data download.

**Contributors** CB, LC, YC and DAvdW contributed to the initial draft and subsequent revisions. CB is the guarantor of the paper. All authors had full access to all of the data and can take responsibility for the integrity of the data and the accuracy of the data analysis. CB affirms that the manuscript is an honest, accurate and transparent account of the study being reported; that no important aspects of the study have been omitted and that any discrepancies from the study as planned have been explained.

**Funding** CB is funded by the National Institute of Health Research School for Primary Care (NIHR SPCR). DAvdW is a member of PROGRESS Medical Research Council Prognosis Research Strategy (PROGRESS) Partnership (G0902393/99558).

**Disclaimer** The views expressed are those of the authors and not necessarily those of the NIHR, the NHS or the Department of health. No other relationships or activities that could appear to have influenced the submitted work.

**Competing interests** None declared.

**Patient consent** Not required.

**Provenance and peer review** Not commissioned; externally peer reviewed.

**Data sharing statement** To ensure patient privacy and confidentiality, data from the CPRD cannot be shared. Therefore, no additional data are available.

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
