## [Reviewer comments · BMJ Open]

ARTICLE DETAILS

TITLE (PROVISIONAL)	Trends in the prevalence, incidence and surgical management of carpal tunnel syndrome between 1993 and 2013: an observational analysis of UK primary care records
AUTHORS	Burton, Claire; Chen, Ying; Chesterton, Linda; van der Windt, Danielle

VERSION 1 – REVIEW

REVIEWER	Dr Jeremy D P Bland East Kent Hospitals University NHS Foundation Trust. Kent, UK
REVIEW RETURNED	26-Oct-2017

GENERAL COMMENTS	This analysis of data from the UK Clinical Practice Research Database provides a valuable longitudinal survey of the perception of carpal tunnel syndrome as seen in primary care over a 20 year period. The authors are well aware of the limitations of their data source: • They are highly dependent on the accuracy of diagnosis in primary care – an environment where access to diagnostic testing is frequently either restricted or not taken up so that the clinical skills of the general practitioner become critical to accurate diagnosis. In general it might be worth adopting the term ‘physician diagnosed CTS’ in several places in this paper to emphasise the dominant case definition applied – we cannot of course tell how many of these cases met any of the more restrictive case definitions. Clinical diagnosis of CTS is clearly not faultless in primary care, nor for that matter even in ordinary orthopaedic practice where a not uncommon reason for failure of surgical treatment is that the diagnosis was wrong. Nevertheless I see no reason to believe that there has been a major and systematic change in the diagnostic acumen of general practitioners with respect to CTS between 1993 and 2013 and these figures therefore probably reflect a consistent sampling of CTS in the source population.• The fact that CTS is most often a bilateral condition, with the non-dominant hand, on average developing symptoms a little after the dominant hand, presents many problems in CTS studies of all kinds. The issue is only addressed rather obliquely in this manuscript but if I understand the authors correctly they have effectively counted only the first presentation with CTS for each individual, and therefore only one operation per individual when counting operations? Potentially therefore there may have been up to twice as many operation performed as reflected in this data and this is perhaps a possible source of differences between hospital episode statistics, which will count two operations in one patient as two episodes, and this data? If I have misunderstood could the authors please make the methods
---

clearer in this respect.

- Some CTS may be completely unknown to primary care. CTS is a known complication of wrist fractures, rheumatoid arthritis, diabetes etc and patients who are attending hospital services for this sort of disorder may have their CTS dealt with in secondary care without anyone ever thinking to inform the GP, or without the information forwarded from secondary care being coded in to the CPRD? Treatment carried out in the private sector (considerable for CTS) may also be missed.

Provided the paper is read with the limitations of the source data in mind I think this provides the best available longitudinal view of CTS in the UK to date and my own speculations on the possible explanations for the trends observed would mostly be very similar to the authors'. In particular the recent decrease in carpal tunnel surgery rates seems highly likely to be related to the widespread adoption of restrictive qualifying criteria for the funding of surgery for CTS by PCTs and CCGs. I have one or two minor suggestions which the authors may wish to consider.

- The introduction contains some material of doubtful relevance such as the statement "Patients with moderate symptoms should be referred..." which seems to be a treatment recommendation, and which in any case fails to define what constitutes 'moderate' symptoms. The next sentences covering surgical options and adjuncts also seems to me to be irrelevant and could be omitted.

- In methods – for those unfamiliar with the CPRD it would be useful to add a sentence or two explaining how representative the practice sampling is geographically and what determines whether practices participate or not – is it a random sampling or is the sample biased towards more 'motivated' practices.

- The third paragraph of 'Methods' on page 5 is rather jargon infested and would benefit from a re-write in to plain English explaining the purpose behind the terms 'up to standard', 'research quality'

- The assumption that a patient with significant CTS symptoms would present to their GP within a period of 2 years from onset may be over-optimistic. I frequently encounter patients who, when asked about the duration of symptoms will admit to a 20 year, sometimes fluctuating, history but who say "It's only been bad enough to bother my GP with it in the last few months", or who have clearly had severe CTS symptoms for several years, self diagnosed as 'a bit of arthritis' before the thenar wasting is noticed by an alert GP. Nevertheless I accept that some rules had to be applied in order to make data extraction from CPRD practicable and the two year criterion is pragmatic – it should just be acknowledged that that is what it is.

- One factor which appears to get relatively little attention in the discussion is the changing age distribution of the UK population. CTS incidence is strongly influenced by age and there is little doubt that the UK population is getting older. The authors tell us that their age-adjusted figures were not greatly different so that they have presented raw data but there clearly have been changes during the study period with the mean age of incidence and mean age at surgery rising across the period (more so in males). It seems very likely that in part, we are seeing more CTS because we have more older people, who also tend to be more likely to have severe CTS, and thus more likely to end up with surgery. It is also notable that the age distribution profiles of CTS in women and men are different with women showing a marked perimenopausal peak in incidence which

	is not seen in men, but both sexes showing a marked increase in incidence from age 65 upwards. If more of the overall CTS population is drawn from the over 65 age group as time passes this would also help to explain the change in male:female ratio with time. Finally, a question, I would be curious to know if the authors can extract from their data the number of interventions per patient for each incident case of CTS over the next 'x' years from the diagnosis (x=5 would be great but one could argue for various time periods) – in particular the number of injections performed, but also the number of operations. Ideally one would want this data for each hand but even overall data per patient would be of great interest. With the increasing pressure from purchasers to treat CTS non-surgically in recent years one might guess that there would be a complementary increase in the use of corticosteroid injection in primary care along with the drop in surgery.
--	---

REVIEWER	Fouquet Santé publique France, the French national public health agency, France
REVIEW RETURNED	04-Jan-2018

GENERAL COMMENTS	This article is interesting. However it is difficult to understand. Introduction: Specify the main individual and occupational risk factors of CTS and specify that this is a major occupational health problem. Methods: The methods part seems to be reorganized. For example, the 2 year period definition for incident CTS is referred in two different parts. In addition, I did not fully understand the difference between the definition of prevalent CTS (which includes a priori CTR) and CTR (the readcodes in Tables 1 and 2 are different for CTR (so finally the CTS do not include CTR?)). Is it "readcodes" or "read codes" ? The statistical methods part is to reorganize also because the joinpoint regression seems to be used only for CTR, whereas this regression is used for prevalent and incident CTS too. The use of the joinpoint seems to be very interesting. Results: The figure are to be reviewed because they are not always the same as those in tables: P7 Line 7: 53, not 42 Line 8: 59, not 48 Line 22: 27.68, not 27.09 Line 23: (95% CI 27.09 - 28.28), not (95% CI 28.28 - 35.95) I would be useful to the sentence "The age and sex standardised estimates of the annual prevalence and incidence of CTS are shown in Supplementary Table 3." after "Unstandardised and standardised rates were very similar, hence we report unstandardised rates as the primary outcome." The reference to Figure 1 should be made before the paragraph Trends in incidence. In a general way about joinpoint regressions, it would be nice to know if they are significant, and if so, what the p-values are. In addition, is there a single p-value for the entire model and / or there are p-values for each phase? In this case, it would be interesting to
--

	indicate them. I also wonder about the fact that the regressions were done only for the entire population. It would have seemed interesting to achieve them by genre since the data is available. Concerning the CTR Similarly, the results by genre are surprising. Contrary to literature, the prevalence rate is higher in men than in women. This result is not discussed in the discussion part whereas it seems paradoxical. In the same way, why do the authors mention the median ages, since they don't commented on? This information could have been useful in particular to discuss hormonal risk factors known to women (pregnancy and menopause). Discussion: The influence of case definition on the results is well discussed. The same is true for potential coding errors and the impact of public policies. However, the evolution of medical and surgical practices is little emphasized. In the same way, it is necessary to discuss the effects of age and sex on CTS. Some results are presented, and not at all commented. And some results (prevalence rate of surgical cases higher in men than women) are not found in Literature. The referencing of supplementary tables 4 and 5 is surprising. This part would require the writing of an entire article of literature review, with precise method. References: References 7, 10 and 30 are to be reviewed. In addition, there are articles on trends of CTS that could have been quoted, for comparison: Mustard et al OEM 2015, Stocks et al OEM 2015, Roquelaure et al SJWEH 2017. Table 5: I don't understand the column "Episodes per 10,000 person years" of this table. It is very surprising that prevalence is higher in men than in women. Figures 1,2,3: What is the sign "^"?
--	--

VERSION 1 – AUTHOR RESPONSE

REVIEWER 1 EVALUATION

This analysis of data from the UK Clinical Practice Research Database provides a valuable longitudinal survey of the perception of carpal tunnel syndrome as seen in primary care over a 20 year period. The authors are well aware of the limitations of their data source:

- They are highly dependent on the accuracy of diagnosis in primary care – an environment where access to diagnostic testing is frequently either restricted or not taken up so that the clinical skills of the general practitioner become critical to accurate diagnosis. In general it might be worth adopting the term 'physician diagnosed CTS' in several places in this paper to emphasise the dominant case definition applied – we cannot of course tell how many of these cases met any of the more restrictive case definitions. Clinical diagnosis of CTS is clearly not faultless in primary care, nor for that matter even in ordinary orthopaedic practice where a not uncommon reason for failure of surgical treatment is that the diagnosis was wrong. Nevertheless I see no reason to believe that there has been a major and systematic change in the diagnostic acumen of general practitioners with respect to CTS between

1993 and 2013 and these figures therefore probably reflect a consistent sampling of CTS in the source population.

We agree with reviewer 1 and unequivocally acknowledge both the limitations and benefits of using consultation data in epidemiology research. The 'Discussion' section describes what in our view are the most substantial of these limitations and the 'Conclusion' section now underlines the case definition of CTS applied in this study, is 'physician diagnosed.' Para3 P 15 clearly emphasises the fact that the paper refers to physician diagnosed CTS.

- The fact that CTS is most often a bilateral condition, with the non-dominant hand, on average developing symptoms a little after the dominant hand, presents many problems in CTS studies of all kinds. The issue is only addressed rather obliquely in this manuscript but if I understand the authors correctly they have effectively counted only the first presentation with CTS for each individual, and therefore only one operation per individual when counting operations? Potentially therefore there may have been up to twice as many operation performed as reflected in this data and this is perhaps a possible source of differences between hospital episode statistics, which will count two operations in one patient as two episodes, and this data? If I have misunderstood could the authors please make the methods clearer in this respect.

The reviewer is correct that only the first recorded episode of surgery is included in the analysis. We found that the Read code for surgery could be used in a patient record multiple times. This is likely due to it being 'cut and pasted' in the consultation software to code follow up clinical interactions. The potential repercussions of this and the fact that contralateral cases cannot be identified in CPRD data are described in para 3 page 17.

- Some CTS may be completely unknown to primary care. CTS is a known complication of wrist fractures, rheumatoid arthritis, diabetes etc and patients who are attending hospital services for this sort of disorder may have their CTS dealt with in secondary care without anyone ever thinking to inform the GP, or without the information forwarded from secondary care being coded in to the CPRD? Treatment carried out in the private sector (considerable for CTS) may also be missed. We acknowledge that it is possible that patients with symptoms of CTS either do not present at all to a primary care physician or receive care outside of the primary care environment. We would still however expect episodes of surgery in the public and private sector to be communicated to primary care and coded appropriately in the patient record.

Provided the paper is read with the limitations of the source data in mind I think this provides the best available longitudinal view of CTS in the UK to date and my own speculations on the possible explanations for the trends observed would mostly be very similar to the authors'. In particular the recent decrease in carpal tunnel surgery rates seems highly likely to be related to the widespread adoption of restrictive qualifying criteria for the funding of surgery for CTS by, PCTs and CCGs . I have one or two minor suggestions which the authors may wish to consider.

- The introduction contains some material of doubtful relevance such as the statement "Patients with moderate symptoms should be referred..." which seems to be a treatment recommendation, and which in any case fails to define what constitutes 'moderate' symptoms. The next sentences covering surgical options and adjuncts also seems to me to be irrelevant and could be omitted.

We feel that a brief description of the treatment pathway of CTS is important in order to contextualise the study. The descriptors of 'mild, moderate and severe' can be found in the referenced source. We have slightly amended the sentence regarding referrals to clarify this a recommendation from a published clinical pathway. The detail regarding surgical adjuncts has been removed.

- In methods – for those unfamiliar with the CPRD it would be useful to add a sentence or two explaining how representative the practice sampling is geographically and what determines whether

practices participate or not – is it a random sampling or is the sample biased towards more ‘motivated’ practices.

Para 1 P 5 describes that the population represented in CPRD has been found to be generalizable to the UK population. A further sentence underlining the fact that the participating practices may not be representative of practices nationwide in terms of size and geography, has been added. To our knowledge the suggestion that practices included in CPRD are more motivated in the quality of care they provide than those that are not, has not been described in the literature.

- The third paragraph of ‘Methods’ on page 5 is rather jargon infested and would benefit from a re-write in to plain English explaining the purpose behind the terms ‘up to standard’, ‘research quality’ The terms ‘up to standard’ and ‘acceptable patient’ data are defined by CPRD and are used to identify data that is of suitable quality for use in research. The paragraph has been amended to clarify the use of these terms.

- The assumption that a patient with significant CTS symptoms would present to their GP within a period of 2 years from onset may be over-optimistic. I frequently encounter patients who, when asked about the duration of symptoms will admit to a 20 year, sometimes fluctuating, history but who say “It’s only been bad enough to bother my GP with it in the last few months”, or who have clearly had severe CTS symptoms for several years, self-diagnosed as ‘a bit of arthritis’ before the thenar wasting is noticed by an alert GP. Nevertheless I accept that some rules had to be applied in order to make data extraction from CPRD practicable and the two year criterion is pragmatic – it should just be acknowledged that that is what it is.

Para 1 P 6 explains the reasons and methods used to define this cut off (needed to define incident cases). A sentence has been added to clarify the fact that this is an assumption based on expert consensus.

- One factor which appears to get relatively little attention in the discussion is the changing age distribution of the UK population. CTS incidence is strongly influenced by age and there is little doubt that the UK population is getting older. The authors tell us that their age-adjusted figures were not greatly different so that they have presented raw data but there clearly have been changes during the study period with the mean age of incidence and mean age at surgery rising across the period (more so in males). It seems very likely that in part, we are seeing more CTS because we have more older people, who also tend to be more likely to have severe CTS, and thus more likely to end up with surgery. It is also notable that the age distribution profiles of CTS in women and men are different with women showing a marked perimenopausal peak in incidence which is not seen in men, but both sexes showing a marked increase in incidence from age 65 upwards. If more of the overall CTS population is drawn from the over 65 age group as time passes this would also help to explain the change in male: female ratio with time.

The age and gender standardised figures are provided in Suppl. table 3 and demonstrate very similar trends to that of the crude data. The full age gender specific prevalence and incidence data and accompanying graphs have been added as supplementary files (tables 3 and 4, figures 1 and 2). Reference to these tables has been added to the results section (Para 2 P7 and Para 1 P10) and discussed in Para 2 P 15.

Finally, a question, I would be curious to know if the authors can extract from their data the number of interventions per patient for each incident case of CTS over the next ‘x’ years from the diagnosis (x=5 would be great but one could argue for various time periods) – in particular the number of injections performed, but also the number of operations. Ideally one would want this data for each hand but even overall data per patient would be of great interest. With the increasing pressure from purchasers to treat CTS non-surgically in recent years one might guess that there would be a complementary increase in the use of corticosteroid injection in primary care along with the drop in surgery.

Whilst we agree this level of information would be of great use in describing the healthcare use of patients with CTS and how this might have changed over time, it is not possible to use these data to answer this question. We know from pilot work that once a term (e.g. carpal tunnel release) has been used once, it may be used multiple (over 20 in some cases) times. This is due to the fact practitioners are offered by their software, previously used terms when coding their patient interaction. This means a patient presenting for sickness certification, post operative problems are likely to be attributed this code. We are also aware that episodes of injection are poorly recorded / detected in the data. Despite using Read codes and linked prescription data, only around 10% of patients had an episode of carpal tunnel injection, which was felt to be low, considering most patients receiving surgery are likely to have received an injection prior to their operation. A full medical record review or prospective observational cohort study would be required to investigate this proposed research further.

REVIEWER 2 EVALUATION

This article is interesting. However it is difficult to understand.

Introduction:

Specify the main individual and occupational risk factors of CTS and specify that this is a major occupational health problem.

Further detail about some of the associations of CTS have been added to the first paragraph of the introduction. This article focuses on CTS occurring in the general population rather than specific occupational groups, which are not recorded in CPRD. Associations with occupation have now been alluded to.

Methods:

The methods part seems to be reorganized. For example, the 2 year period definition for incident CTS is referred in two different parts. In addition, I did not fully understand the difference between the definition of prevalent CTS (which includes a priori CTR) and CTR (the readcodes in Tables 1 and 2 are different for CTR (so finally the CTS do not include CTR?)).

Paragraph 1 Page 6 describes the identification of both the numerator and denominator populations for the purposes of calculating incidence, hence why the 2 year period is mentioned twice. Para 3 P 6 explains why the Read codes for the definition of prevalence and the identification of a surgical episode are slightly different. Surgical revisions are not included in prevalence calculations, as they may be representative of iatrogenic CTS. Other than this small proportion of codes, if a patient had been attributed a treatment code (surgery or injection) but not a diagnostic code, they were included as a prevalent / incident case.

Is it "readcodes" or "read codes" ?

It should read 'Read codes.' Inconsistencies have been amended.

The statistical methods part is to reorganize also because the joinpoint regression seems to be used only for CTR, whereas this regression is used for prevalent and incident CTS too.

The use of the joinpoint seems to be very interesting.

Jointpoint regression has been used to illustrate the trends in the prevalence and incidence as well as surgery (see figures 1 and 2). We have clarified this in the statistical methods (page 7).

Results:

The figure are to be reviewed because they are not always the same as those in tables:

P7

Line 7: 53, not 42

Line 8: 59, not 48

Line 22: 27.68, not 27.09

Line 23: (95% CI 27.09 - 28.28), not (95% CI 28.28 - 35.95)
These figures have been checked and amended accordingly.

I would be useful to the sentence "The age and sex standardised estimates of the annual prevalence and incidence of CTS are shown in Supplementary Table 3." after "Unstandardised and standardised rates were very similar, hence we report unstandardised rates as the primary outcome."
This has been added for further clarity.

The reference to Figure 1 should be made before the paragraph Trends in incidence.
Reference to Figure 1 is made in the paragraph 'Trends in prevalence.'

In a general way about joinpoint regressions, it would be nice to know if they are significant, and if so, what the p-values are. In addition, is there a single p-value for the entire model and / or there are p-values for each phase? In this case, it would be interesting to indicate them.
The Joinpoint programme selects the best fitting model to describe the trend over time. There is not one overall p value. The segments marked ^ means that the annual percentage change is significantly different from zero, with a significance level $p < 0.05$. The tables for each model has been added to illustrate these p values.

I also wonder about the fact that the regressions were done only for the entire population. It would have seemed interesting to achieve them by genre since the data is available.
Whilst this would be possible in theory, we know from standardising the data to the population structure of 2013, that the observed trends are the same. Producing multiple further outputs would not therefore alter the overall conclusion. Suppl. Figures 1 and 3 have been added to provide an illustration of prevalence and incidence by age and gender.

Concerning the CTR Similarly, the results by genre are surprising. Contrary to literature, the prevalence rate is higher in men than in women. This result is not discussed in the discussion part whereas it seems paradoxical.
At no time do we suggest that more men are receiving surgery than women. The data we present and discuss pertains to the proportion of men and women with CTS who receive surgical treatment. A greater proportion of men with prevalent CTS have surgery, when compared to women.

In the same way, why do the authors mention the median ages, since they don't commented on? This information could have been useful in particular to discuss hormonal risk factors known to women (pregnancy and menopause).
The median ages in the prevalent, incident and surgical patient groups have been commented on in each corresponding paragraph of the results section. Further mention of these ages has been added to the first paragraph of the discussion section.

Discussion:

The influence of case definition on the results is well discussed. The same is true for potential coding errors and the impact of public policies. However, the evolution of medical and surgical practices is little emphasized. In the same way, it is necessary to discuss the effects of age and sex on CTS. Some results are presented, and not at all commented. And some results (prevalence rate of surgical cases higher in men than women) are not found in Literature.
Further detail regarding the age and gender of prevalent patients has been added to the discussion section (Para 1). To our knowledge the surgical methods have not changed substantially over the past 20 years, however, the way they are delivered in the national healthcare system has, which is what has been discussed at length. All results are now commented on and brought into the discussion, however age and gender do not require particular attention as we know that standardising the data did not make any substantial difference to the results presented (hence the crude data is

provided with the standardised data included as a supplementary file). We do not suggest the prevalence of surgery is greater in the male population, rather, a greater proportion of men with prevalent CTS have surgery.

The referencing of supplementary tables 4 and 5 is surprising. This part would require the writing of an entire article of literature review, with precise method.

Tables 4 and 5 do not attempt to represent a formalised systematic review of the literature. They are a summary of data found using rapid review methods. A phrase in the discussion section Para 1 has been added to this effect.

References:

References 7, 10 and 30 are to be reviewed.

In addition, there are articles on trends of CTS that could have been quoted, for comparison: Mustard et al OEM 2015, Stocks et al OEM 2015, Roquelaure et al SJWEH 2017.

The articles included in tables 4 and 5 are particular to the general population in order that their results could be compared to the results of this study. The above suggestions are gratefully noted but unfortunately focus on CTS in specific work related populations and are hence not comparable to these results.

Table 5:

I don't understand the column "Episodes per 10,000 person years" of this table. It is very surprising that prevalence is higher in men than in women.

The column 'episodes per 10,000 person years' relates to the raw numbers of surgical episodes per 10,000 person years where as the following data presented reflects the proportion of male and female patients with CTS who have a surgical episode. We do not suggest that the prevalence of surgery is higher in men.

Figures 1,2,3:

What is the sign "^"?

The legends have been amended to explain that ^ reflects significance at the 0.05 level.

Many thanks for your further consideration of this article for publication.

VERSION 2 – REVIEW

REVIEWER	Jeremy Bland East Kent Hospitals Univeristy NHS Foundation Trust
REVIEW RETURNED	10-Feb-2018

GENERAL COMMENTS	I am fairly happy with the revisions to this article. I still think that the third paragraph of Methods, relating to the 'up to standard' and 'acceptable patient' terms, is clumsy. Would it not be easier to dispense with these terms, which are derived from the technical description of the CPRD, and simply say "Data was only used from practices which met a data quality standard based on continuity of recorded data, and from patients who had a record including at least their status, age and gender" (though I would like to know quite what the patient 'status' means)? New data is now included showing the age/sex specific incidence, Suppl Figs 2 and 4. Interestingly these figures closely match my own data published in 2003 but the description of this in the text is slightly
---

	misleading (1st paragraph of results), describing the pattern in females as 'peaks in the 50-59 age group' - implying a unimodal distribution. In fact the distribution in females is bimodal and this can be seen in the authors data in the higher incidence in the 70+ age group than in the 60-69 age group. In fact both males and females show the marked peak in incidence in the elderly, which is presumably age related, whereas it is only females who show a pronounce peak in middle age which one assumes is peri-menopausal.
REVIEWER	Fouquet Santé publique France, France
REVIEW RETURNED	02-Mar-2018
GENERAL COMMENTS	The authors have sufficiently responded to, and addressed all previous comments. No further comments.

VERSION 2 – AUTHOR RESPONSE

REVIEWER 1 EVALUATION

I am fairly happy with the revisions to this article. I still think that the third paragraph of Methods, relating to the 'up to standard' and 'acceptable patient' terms, is clumsy. Would it not be easier to dispense with these terms, which are derived from the technical description of the CPRD, and simply say "Data was only used from practices which met a data quality standard based on continuity of recorded data, and from patients who had a record including at least their status, age and gender" (though I would like to know quite what the patient 'status' means)?

As recommended by the reviewer, the technical terms used in the aforementioned paragraphs have been removed and the suggested explanation used instead. For information, status refers to the registration status of a patient (registered, unregistered, temporarily registered etc).

New data is now included showing the age/sex specific incidence, Suppl Figs 2 and 4. Interestingly these figures closely match my own data published in 2003 but the description of this in the text is slightly misleading (1st paragraph of results), describing the pattern in females as 'peaks in the 50-59 age group' - implying a unimodal distribution. In fact the distribution in females is bimodal and this can be seen in the authors data in the higher incidence in the 70+ age group than in the 60-69 age group. In fact both males and females show the marked peak in incidence in the elderly, which is presumably age related, whereas it is only females who show a pronounce peak in middle age which one assumes is peri-menopausal.

As suggested, the first and second paragraphs in the results section have been expanded to describe more fully the supplementary data provided.